# Purpura Fulminans in an Extremely Premature Infant: A Case Report

**DOI:** 10.3390/children12111546

**Published:** 2025-11-15

**Authors:** Anna Rojas Roig, Eduardo Costa Félix de Oliveira, Cristina Borràs-Novell, Anna Álvarez Martínez, Ana Espinosa Gimenez, Miguel Bejarano Serrano, Gemma Pérez Acevedo, Carmen Gracia, Àfrica Pertierra Cortada, Miguel Alsina Casanova

**Affiliations:** 1Neonatology Department, BCNatal—Centre de Medicina Maternofetal i Neonatologia de Barcelona, Hospital Clínic, Universitat de Barcelona, 08028 Barcelona, Spain; 2Neonatology Department, BCNatal—Centre de Medicina Maternofetal i Neonatologia de Barcelona, Hospital Sant Joan de Déu, Universitat de Barcelona, 08950 Barcelona, Spain; 3Pediatric Surgery Department, Hospital Sant Joan de Déu, Universitat de Barcelona, 08950 Barcelona, Spain; 4Pediatric Nurse, Hospital Sant Joan de Déu, Universitat de Barcelona, 08950 Barcelona, Spain

**Keywords:** purpura fulminans, blood coagulation disorders, premature infant, prematurity, case report

## Abstract

**Highlights:**

**What are the main findings?**
Purpura fulminans can occur in extremely premature infants, although rarely described.Early recognition and aggressive multidisciplinary treatment are essential for survival.

**What is the implication of the main finding?**
Ethical decision-making and family-centered care are key throughout the clinical course.

**Abstract:**

Neonatal purpura fulminans is a rare and challenging diagnosis due to its resemblance to other necrotizing skin conditions and the immature coagulation system in newborns. Early multidisciplinary intervention is key. We present the case of an extremely premature infant, born at 24 + 3 weeks’ gestation and weighing 520 g. Clinical evolution, diagnostic approach, and therapeutic strategies are described. By day 5, the infant developed hemorrhagic-necrotic skin lesions. Diagnosis of purpura fulminans led to broad-spectrum antibiotics, anticoagulation, supportive care, and surgery. Despite complications such as osteomyelitis and scarring, the patient’s condition improved. Genetic testing ruled out congenital protein C/S deficiency, suggesting an infectious etiology. Therapeutic decisions were guided by ethical considerations, prioritizing family-centered care and patient comfort. This case adds to the limited literature on purpura fulminans in preterm infants and, to our knowledge, represents the smallest patient reported to date.

## 1. Introduction

Purpura fulminans is a rare, life-threatening thrombotic disorder characterized by the sudden onset of hemorrhagic-necrotic skin lesions, disseminated intravascular coagulation, and vascular thrombosis. It is associated with congenital deficiencies of protein C or S or acquired conditions such as infections. In neonates, purpura fulminans poses a diagnostic and therapeutic challenge due to the overlap with other necrotizing skin conditions and the physiological immaturity of the coagulation system. Early recognition and aggressive multidisciplinary management are essential to improve outcomes, as the disease can rapidly progress to multiorgan failure and death.

This case report describes the clinical course of purpura fulminans in an extremely premature infant born at 24 + 3 weeks of gestation, who, to our knowledge, is the smallest reported to date. This case highlights the diagnostic complexity, the importance of a broad differential diagnosis, and the role of advanced wound care and ethical decision-making in neonatal intensive care. The favourable clinical evolution observed, despite the severity of complications, highlights the potential for recovery when multidisciplinary interventions are implemented. This case offers valuable insights into the understanding and management of purpura fulminans in highly vulnerable neonatal populations.

## 2. Case Report

We report the case of an extremely premature newborn (24 + 3 weeks’ of gestation) with a birth weight appropriate for gestational age (520 g, 25th percentile), resulting from an in vitro fertilization pregnancy with oocyte donation from the non-gestational progenitor. The pregnancy was well-monitored, with normal ultrasound findings and negative maternal serologic tests. Relevant maternal history included a hemorrhagic stroke of unknown etiology at 12 weeks of gestation and cervical incompetence, which was managed with a pessary.

The mother presented with metrorrhagia and uterine contractions, ultimately leading to an urgent cesarean section due to breech presentation, after the administration of a full course of antenatal corticosteroid therapy and neuroprotection with magnesium sulfate. At birth, the infant’s APGAR scores were 6, 7, and 9, and required intermittent positive pressure ventilation for 10 min before transitioning to non-invasive ventilation. Antihemorrhagic prophylaxis was administered with 1 mg of intravenous vitamin K. Umbilical venous and arterial catheters were placed. Due to clinical and ultrasonographic evidence of neonatal respiratory distress syndrome, a dose of tracheal surfactant was administered via the Less Invasive Surfactant Administration (LISA) technique at 2 h of life. At 18 h of life, the patient’s respiratory status deteriorated, requiring endotracheal intubation and a second dose of surfactant. In light of the potential for early sepsis, blood cultures were collected, and antibiotic therapy with ampicillin and cefotaxime was started immediately after birth, along with antifungal prophylaxis with fluconazole. The initial laboratory tests showed a white blood cell count of 5760/μL, with 71% lymphocytes and 16% neutrophils, a C-reactive protein (CRP) level of 0.02 mg/dL, and a platelet count of 231,000/μL. On the second day of life, a transfontanellar ultrasound revealed bilateral grade II intraventricular hemorrhage.

On day 5 of life, the patient developed rapidly progressive hemorrhagic-necrotic lesions involving a large portion of the trunk, proximal lower extremities, perineal, and gluteal areas, despite ongoing systemic antibiotic therapy and daily topical wound care (Figure 1). The patient required inotropic support for 48 h due to hypovolemia secondary to anasarca, accompanied by hypoalbuminemia and prerenal acute kidney injury.

Laboratory tests revealed elevated acute phase reactants (maximum CRP: 267 mg/L; maximum PCT: 6.97 ng/mL), leukopenia (1830/μL) with neutropenia (330/μL), normocytic anemia (Hemoglobin 10.6 g/dL, mean corpuscular volume 114.2 fL), and thrombocytopenia (23,000/μL) requiring multiple transfusions, without associated coagulopathy. The prothrombin time ranged between 69.2% and 76.4%, corresponding to 12.9–13.7 s. The activated partial thromboplastin time ranged between 32.3 and 34 s. Fibrinogen levels were within normal limits, ranging from 3.8 to 5.2 g/L. Serial microbiological assessments were performed, with *Staphylococcus haemolyticus* isolated from the blood culture obtained on day 5 of hospitalization. Serial echocardiograms, abdominal ultrasounds, and transfontanellar scans showed no evidence of internal thrombosis.

Following a multidisciplinary evaluation, the patient was diagnosed with neonatal purpura fulminans. Empiric broad-spectrum antimicrobial therapy with meropenem, vancomycin, and amphotericin B was started after obtaining blood cultures and conducting a serum galactomannan assay, the latter of which returned negative. Adjunctive treatment included vitamin K, fresh frozen plasma, and low-molecular-weight heparin at a dose of 2 mg/kg. Despite these, the skin lesions progressed over the next 48 h, prompting escalation of antimicrobial therapy to include daptomycin and voriconazole. Daily topical wound care was performed by the nursing team. Initially, dressings were performed using Linitul, later transitioning to Microdacyn soaks, followed by drying and application of dressings. Topical fusidic acid was also used in ointment and cream form. Given the clinical situation, the patient required multiple transfusions of blood products and high-dose opioid sedoanalgesia, achieving adequate pain control.

By day 14 of life, the skin lesions had stabilized, acute phase reactants had decreased (C-reactive protein: 8.7 mg/dL; procalcitonin: 1.02 ng/mL), and leukocyte counts had recovered (white blood cell count: 17,860/μL). In addition, transfusion requirements had significantly decreased. Treatment was gradually de-escalated: daptomycin and voriconazole were discontinued after 10 and 5 days, respectively, while meropenem and vancomycin were continued for a total of 18 days. Fresh frozen plasma was administered until 19 days of life, and low-molecular-weight heparin was maintained until genetic testing confirmed the absence of protein C/S deficiency, with normal protein C and S levels after discontinuation.

The necrotizing lesions progressed to desquamation, with areas of full-thickness tissue loss exposing the underlying muscular fascia. At 36 days of life, exposure of rib arches was noted on the right thorax. Extensive debridement was performed, including the exposed ribs, which appeared devascularized and completely detached. Vacuum-assisted closure therapy was initiated. Given the extent of the injury, a bovine-synthetic dermal regeneration template was applied at 64 days of life. The lesions evolved with a tendency toward hypertrophic scarring and cutaneous contractures. The patient developed graft infection with associated costal osteomyelitis, requiring antibiotic therapy with piperacillin/tazobactam for 3 months. The lesions improved, but with a tendency toward hypertrophic scarring and cutaneous contractures, predominantly on the dorsal side (Figure 2).

Due to the prematurity and the severity of the clinical condition, the patient required parenteral nutrition until the 35th day of life, which was associated with cholestasis. She also had a patent ductus arteriosus requiring therapeutic catheterization due to persistence after pharmacological treatment with acetaminophen, as ibuprofen was contraindicated. The patient developed bronchopulmonary dysplasia and a restrictive pulmonary pattern secondary to costal scarring and retractions. Repeated extubation failures ultimately led to a tracheostomy. Magnetic resonance imaging of the brain, performed at term-corrected age, revealed delayed myelination. The patient shows psychomotor developmental delay and growth retardation under multidisciplinary follow-up.

Throughout the clinical course, the multidisciplinary clinical team regularly reassessed the ethical appropriateness of continuing therapeutic measures in collaboration with the family. Given the absence of signs suggestive of severe neurological injury and the patient’s stability despite complex treatment, a decision was made to maintain a proactive management strategy.

## 3. Discussion

This report describes a 24-week gestation neonate diagnosed with purpura fulminans, likely triggered by an infection, who required a complex multidisciplinary management and represented a significant ethical challenge. Few cases of purpura fulminans in preterm infants have been described [1,2,3,4,5]; to our knowledge, this patient is the smallest reported to date.

Purpura fulminans is a rapidly progressing acute prothrombotic disorder due to a protein C or S deficiency, characterized by dermal microvasculature thrombosis and skin necrosis. It can involve large-vessel thrombosis and constitutes a hematological emergency. Clinically, it presents with erythematous macules evolving into rapidly progressive hemorrhagic-necrotic plaques, which can have different distribution patterns depending on the etiology [3,6,7,8,9].

Neonatal purpura fulminans typically begins in areas of trauma and can present with central venous thrombosis and periventricular hemorrhagic infarctions, posing a risk of obstructive hydrocephalus and ocular involvement due to retinal vessel thrombosis [5,6,7,8].

Laboratory findings are often characterized by the presence of features of disseminated intravascular coagulation, such as thrombocytopenia, consumptive hypofibrinogenemia, coagulopathy (prolonged prothrombin and activated partial thromboplastin time), and elevated D-dimer levels [5,7,8,10]. The underlying protein C or S deficiency can be due to congenital or, more commonly, acquired etiologies (i.e., sepsis, post-infectious autoimmunity, anticoagulant therapy, liver disease, galactosemia, severe congenital heart disease, etc.) [5,7,8,10]. Therefore, assessing the serum levels of protein C and S in the patient before initiating treatment, as well as quantifying them in both progenitors, can be helpful in the diagnosis [4]. However, it is important to note that healthy neonates may naturally exhibit hemostatic immaturity with decreased levels of protein C and S, along with interindividual variability. Furthermore, in cases of prematurity, early respiratory distress or clinical severity, protein C or S deficiency may be exacerbated. However, this does not necessarily correlate with the development of purpura fulminans, and normal reference values for this population are not well established. In such cases, genetic testing to exclude congenital protein C or S deficiency is warranted [6,7].

The present case exhibited skin lesions suggestive of purpura fulminans, and a thorough workup did not find evidence of large-vessel thrombosis. Laboratory findings revealed thrombocytopenia with normal coagulation parameters. The differential diagnoses include purpura fulminans (of congenital or infectious origin), necrotizing fasciitis and systemic fungal infection [7]. In the latter two conditions, similar lesions can occur. However, necrotizing fasciitis typically progresses faster and is associated with severe systemic involvement. Although the additional tests were negative, empiric antifungal treatment was administered.

Given the characteristic skin lesions and laboratory findings, a diagnosis of neonatal PF was made, and the described treatment was initiated. Protein C and S levels were not measured before starting treatment. Genetic testing was therefore requested, which turned out negative. Although no causative microorganism was identified, the clinical and analytical improvement following broad-spectrum antimicrobial therapy and supportive care supports an infectious etiology.

Given the broad and multifactorial etiology of neonatal purpura fulminans, a structured summary can help clarify the main contributing factors and their clinical implications. Table 1 provides an overview of the clinical manifestations, underlying causes (genetic and acquired), and major complications reported in the literature.

As shown, both congenital and acquired mechanisms can lead to similar clinical presentations, though the underlying pathophysiology and management implications differ substantially. In the present case, the absence of a genetic mutation, together with a favorable response to antimicrobial and supportive therapy, supports an infection-related etiology.

In the case reported by Aygun et al., purpura fulminans developed in a premature infant due to Stenotrophomonas maltophilia infection. The presentation included rapidly progressive skin necrosis and systemic deterioration, and despite aggressive antimicrobial and supportive therapy, the outcome was fatal [1]. Similarly, Teertstra et al. described purpura fulminans secondary to Serratia marcescens septicemia in a preterm neonate, with a comparable clinical course and poor outcome [5]. In contrast, Elayappen et al. reported purpura fulminans in a premature twin associated with late-onset group B streptococcal sepsis. The infant presented with hemorrhagic skin lesions and coagulopathy, received prompt antibiotic and supportive treatment, and survived without major sequelae [2]. Our patient developed extensive hemorrhagic-necrotic skin lesions by day 5 of life, attributed to a probable infectious etiology. Management included broad-spectrum antibiotics, anticoagulation, transfusions, advanced wound care, and surgery. Genetic testing excluded congenital protein C/S deficiency, supporting an acquired etiology. Despite the severity and multiple complications, the patient survived, demonstrating that favorable outcomes are possible in extremely premature infants with timely, multidisciplinary, and ethically guided care.

Regardless of the etiology, purpura fulminans is associated with high morbidity and mortality, frequently progressing to multiorgan failure and representing a life-threatening condition. Therefore, treatment must be initiated immediately upon suspicion [7,8,11]. A potential septic condition must be assumed, with early initiation of broad-spectrum antimicrobial therapy, volume expansion, and inotropes and/or corticosteroids if necessary. If analytical findings suggest disseminated intravascular coagulation, fresh frozen plasma at 10–20 mL/kg every 6–12 h should be administered to correct the underlying coagulation factor and/or protein C or S deficiencies [12,13]. Additionally, supportive treatment with fluid therapy, blood transfusions, unfractionated heparin to halt thrombosis, topical wound care, sedation and analgesia, and surgical intervention if required should be provided [2,5]. If congenital protein C deficiency is confirmed, protein C concentrate (Ceprotin, Protexel^®^) should replace fresh frozen plasma due to its greater efficacy and fewer adverse effects [2,5,8].

Skin sequelae will depend on the extent of necrosis, ranging from near-complete healing to the need for amputation, debridement or fasciotomy. In cases of purpura fulminans secondary to sepsis or post-infectious, controlling the underlying disease should stop the development of new lesions. In cases of hereditary protein C deficiency, prophylaxis with fresh frozen plasma or protein C/S administration, along with anticoagulation, will minimize the risk of recurrences. However, the only curative treatment remains liver transplantation [7].

Shared decision-making and parental involvement were integral to the clinical management of this extremely premature infant with purpura fulminans. The medical team conducted regular ethical evaluations and engaged the family in transparent discussions regarding prognosis, treatment options, and anticipated outcomes. This collaborative approach ensured that care decisions reflected both clinical judgment and the family’s values, fostering a compassionate and ethically grounded framework. Initially, limited therapeutic measures were adopted due to the severity of the condition and poor early response. However, as the patient’s status improved, the family’s commitment to active care influenced the escalation of treatment, including advanced wound management and long-term supportive therapies.

## 4. Conclusions

We presented the case of an extremely premature newborn diagnosed with purpura fulminans, likely triggered by an infectious episode without an identified pathogen. After ruling out congenital protein C and/or S deficiency, a more favorable prognosis was anticipated, with low risk of recurrence and no need for long-term anticoagulant therapy. Despite this, the patient experienced significant morbidity due to purpura fulminans and complications related to extreme prematurity, which posed ethical and therapeutic challenges. Initially, treatment options were limited due to poor clinical response, but following clinical improvement and in alignment with the family’s wishes, care was optimized while prioritizing the patient’s comfort and family well-being.

## Figures and Tables

**Figure 1 children-12-01546-f001:**
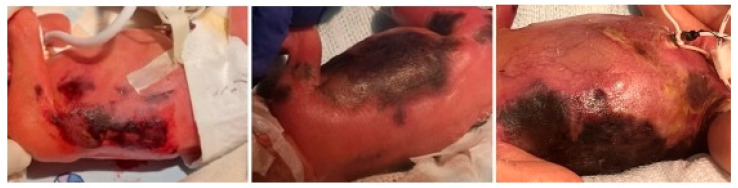
Progression of hemorrhagic-necrotic lesions on the right lateral trunk from 5 to 14 days of life.

**Figure 2 children-12-01546-f002:**
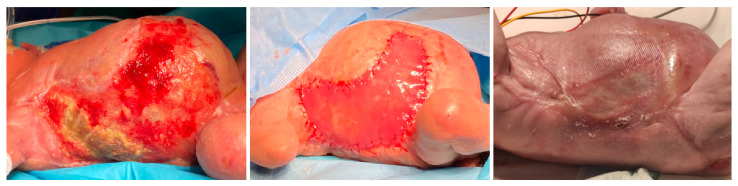
Evolution of skin lesions on the right lateral trunk and abdomen from 36 to 64 days of life. Rib cage exposure is visible in the first image. Placement of Dermal Template is visible in the second image. Evidence of hypertrophic scarring tendency is visible in the third image.

**Table 1 children-12-01546-t001:** Overview of the clinical manifestations, underlying causes (genetic and acquired), and major complications of neonatal purpura fulminans.

Category	Subtype/Feature	Examples/Details	References
Clinical Manifestations	Purpura, necrosis, gangrene	Well-demarcated purpura, skin necrosis, eschars	[1,2,3]
Etiology	Genetic causes	Congenital protein C/S deficiency, complement variants	[4,5,6,7,8]
	Acquired causes	Sepsis (bacterial, viral), DIC, catheter-related	[1,3,4,8,9,10,11]
Complications	Circulatory compromise	Limb ischemia, multi-organ failure, amputation	[2,3,5,10]
	Hemorrhagic complications	Intracranial, retinal, GI hemorrhage	[5,7]
	Long-term sequelae	Scarring, tissue loss, need for lifelong therapy	[2,4,5,8]

## Data Availability

The data supporting the findings of this case report are not publicly available due to patient privacy concerns, but de-identified data may be available from the corresponding author upon reasonable request.

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
