# Peer review of "Purpura Fulminans in an Extremely Premature Infant: A Case Report"

_children, 2025, doi:10.3390/children12111546_

Round 1
Reviewer 1 Report
Comments and Suggestions for Authors
Thank you for giving me the opportunity to read this article. The diagnosis of purpura fulminans in a very young premature baby is heavily discussed, and it actually appears to be a complication of DIC and septic embolism in a sepsis case. I found the practical and clinical aspects of the article very well-written, and the patient management results were very interesting and led to positive results. However, from a scientific perspective, discussing purpura fulminans as a condition with no genetic basis or cause seemed a bit oversimplified to me. Catheters, sepsis, and the newborn's own coagulation potential and inadequacy should be discussed more prominently in neonatal intensive care, and I believe a small table or figure would be much more useful. I suggest that the etiology be presented separately in a table with purpura, necrosis, or circulatory compromise under the heading, and clinical causes, genetic causes, and complications under the heading.
Author Response
We have carefully reviewed the comments and made the following improvements:

Reviewer 2 Report
Comments and Suggestions for Authors
Please mention whether the first surfactant was given by LISA. Please describe what did you use for topical wound care. What antibiotics did you use for 3 months during osteomyelitis treatment. Was PICC line or central line used for parenteral nutrition. Please mention what was the outcome of the baby.
This is a case report of Purpura Fulminans in a extreme preterm neonate(24 weeks+3 days). There are many cases in literature but this is in extreme preterm. It adds that extreme preterm with purpura fulminans can be managed, however outcome of the baby is not mentioned anywhere in the script. Hence we would like to know the outcome of the baby. References are appropriate. Methodology is not important here because it is a case report.
Author Response

(The authors gave the same response as above.)

Reviewer 3 Report
Comments and Suggestions for Authors
This case report describes purpura fulminans (PF) in an extremely premature infant born at 24 + 3/7 weeks. The major limitation is that the diagnosis of PF remains somewhat uncertain. I have the following comments:
- L67–68: Please clarify what “complete fetal lung maturation” means. Was antenatal corticosteroid therapy administered before birth?
- L75–77: What was the timing of antibiotic initiation—immediately after birth or after intubation? Please specify.
- The initial laboratory data should be presented to allow comparison with those obtained during the septic period.
- L89–90: Please provide the exact laboratory values for leukopenia with neutropenia, normocytic anemia, and thrombocytopenia. In addition, the DIC profile (PT, aPTT, fibrinogen, D-dimer) should be included, as these values are critical for confirming the diagnosis of purpura fulminans.
- L92: Please clarify whether Staphylococcus haemolyticus grew in the first set of blood cultures or during the sepsis episode. The term “initial” is vague.
- L105–106: Specify the exact laboratory test results mentioned in this section.
- The Conclusion section is too long. Please condense it into a concise summary emphasizing the key teaching points in 3–5 sentences.
- Suggest comparing this case with previously reported preterm PF cases in the Discussion to highlight similarities and differences in presentation, management, and outcomes.
- The major limitation of this manuscript is the lack of a definitive diagnosis. No causative pathogen was identified, and the absence of typical DIC findings (PT/aPTT, fibrinogen) is atypical for PF and requires further clarification. This limitation substantially affects the overall interpretive strength of the case.
- The authors emphasize the importance of ethical considerations, but the manuscript lacks detail on the decision-making process with the family. Please elaborate on how shared decision-making and parental involvement influenced clinical management.
Author Response

(The authors gave the same response as above.)

Round 2
Reviewer 1 Report
Comments and Suggestions for Authors
I advice to accept this last form
Thank you
Reviewer 3 Report
Comments and Suggestions for Authors
The authors addressed the issues I mentioned. I have no further comments.